# *Christensenella minuta* Alleviates Acetaminophen-Induced Hepatotoxicity by Regulating Phenylalanine Metabolism

**DOI:** 10.3390/nu16142314

**Published:** 2024-07-18

**Authors:** Ting Yao, Liyun Fu, Youhe Wu, Lanjuan Li

**Affiliations:** State Key Laboratory for Diagnosis and Treatment of Infectious Diseases, National Clinical Research Center for Infectious Diseases, National Medical Center for Infectious Diseases, Collaborative Innovation Center for Diagnosis and Treatment of Infectious Diseases, The First Affiliated Hospital, Zhejiang University School of Medicine, 79 Qingchun Rd., Hangzhou City 310003, China

**Keywords:** acetaminophen-induced liver injury, *Christensenella minuta*, phenylalanine, dysbiosis, MAPK signaling pathway

## Abstract

Acetaminophen (APAP)-induced liver injury (AILI), even liver failure, is a significant challenge due to the limited availability of therapeutic medicine. *Christensenella minuta* (*C. minuta*)*,* as a probiotic therapy, has shown promising prospects in metabolism and inflammatory diseases. Our research aimed to examine the influence of *C. minuta* on AILI and explore the molecular pathways underlying it. We found that administration of *C. minuta* remarkably alleviated AILI in a mouse model, as evidenced by decreased levels of alanine transaminase (ALT) and aspartate aminotransferase (AST) and improvements in the histopathological features of liver sections. Additionally, there was a notable decrease in malondialdehyde (MDA), accompanied by restoration of the reduced glutathione/oxidized glutathione (GSH/GSSG) balance, and superoxide dismutase (SOD) activity. Furthermore, there was a significant reduction in inflammatory markers (IL6, IL1β, TNF-α). *C. minuta* regulated phenylalanine metabolism. No significant difference in intestinal permeability was observed in either the model group or the treatment group. High levels of phenylalanine aggravated liver damage, which may be linked to phenylalanine-induced dysbiosis and dysregulation in cytochrome P450 metabolism, sphingolipid metabolism, the PI3K-AKT pathway, and the Integrin pathway. Furthermore, *C. minuta* restored the diversity of the microbiota, modulated metabolic pathways and MAPK pathway. Overall, this research demonstrates that supplementing with *C. minuta* offers both preventive and remedial benefits against AILI by modulating the gut microbiota, phenylalanine metabolism, oxidative stress, and the MAPK pathway, with high phenylalanine supplementation being identified as a risk factor exacerbating liver injury.

## 1. Introduction

The incidence rate of drug-induced liver injury (DILI) remains an important global health concern and a primary reason for drug withdrawal from the market. DILI rarely cause death, but can be serious and is “largely unpredictable” for most drugs that do not appear to cause DILI in a dose-dependent fashion [1]. Notably, acetaminophen (APAP), an antipyretic and analgesic agent, frequently induces hepatotoxicity in clinical settings, even leading to acute liver failure. This has culminated in approximately 68,000 annual emergency department visits, 30,000 hospitalizations, and 500 fatalities in the United States alone, with a similarly concerning scenario evident in Europe due to APAP usage [2,3]. *N*-acetyl-*p*-benzoquinone imine (NAPQI) is an oxidative metabolite of APAP, primarily metabolized by cytochrome P450 enzymes [4]. Under physiological conditions, the majority of APAP is sulfonated and glucuronidated in the liver for detoxification, with only a fraction metabolized into NAPQI, which consumes glutathione (GSH), an antioxidant agent [5]. However, in instances of inadequate GSH supply or APAP overdose, excessive NAPQI binds to macromolecules, instigates mitochondrial damage and augments reactive oxygen species (ROS) levels [4]. *N*-acetylcysteine (NAC), with antioxidant ability, has been established as a common therapeutic intervention. Nevertheless, the narrow therapeutic window of NAC only reduces a small portion of mortality rates in APAP toxicity [6]. Therefore, delineation of the optimal therapeutic approach for AILI remains an area warranting further exploration.

Besides oxidative stress within the mitochondria, multiple factors, including endoplasmic reticulum pressure, autophagy, non-infectious inflammation, and liver repairs, have been implicated in the pathogenesis of AILI [6]. Restoration of gut microbiota dysbiosis is recognized as an effective strategy for maintaining host homeostasis. Gut microbiota impact the liver function of the host through the ‘microbiota–gut–liver’ axis, as the liver is the first organ to come into contact with blood from the intestines [7,8]. For instance, supplementation with *Lactobacillus rhamnosus GG* in a mouse model prevented liver fibrosis by suppressing hepatic bile acid production and boosting bile acid elimination [9]. Similarly, *Akkermansia muciniphila* has been shown to ameliorate AILI by modulating gut microbial composition and metabolism [10].

*Christensenella minuta* (*C. minuta*), a novel generation probiotic belonging to the Christensenellaceae bacteria, is renowned for its positive association with excellent metabolic characteristics, such as a comparatively reduced body mass index (BMI) and reduced serum lipids. *C. minuta* has been suggested as a potentially effective probiotic for the treatment of various diseases, such as type 2 diabetes (T2D) by acylating bile acids (BAs), obesity, and hypercholesterolemia, all of which are related to liver function, or by secreting short-chain fatty acids (SCFAs) to resist inflammation in inflammatory bowel disease [11,12,13]. AILI severely causes hepatocyte damage or death, leading to metabolic disorders. There are knowledge gaps regarding the influence of *C. minuta* on AILI.

In this study, we delved into the potential therapeutic impact of *C. minuta* on the process of AILI and assessed how administering *C. minuta* influenced critical molecular markers and pathways associated with liver damage, ranging from oxidative stress to inflammation. Non-targeted metabolomics was utilized to identify differential metabolites (DMs), with subsequent confirmation of the influence of potentially altered critical metabolites on liver injury. Additionally, 16S rRNA analysis of fecal samples was conducted to observe changes in gut microbiota, while mRNA sequencing of liver tissue was performed to explore underlying mechanisms. Ultimately, by integrating these multi-omics data, we aimed to elucidate and identify potential mechanisms linking the gut and liver in AILI. Our study revealed the efficacy of *C. minuta* in ameliorating AILI through regulation of phenylalanine metabolism.

## 2. Materials and Methods

### 2.1. Culture of C. minuta

*C. minuta* (DSM22607) procured from Testobio company (Ningbo, China) underwent cultivation in a modified liquid Gifu Anaerobic Medium (Hopebiol, Qingdao, China) under anaerobic conditions at 37 °C for 72 h. Following cultivation, the bacteria were centrifugated and resuspended to a concentration of 10^9^ CFU/mL in a 16% glycerol solution.

### 2.2. Animals and Experimental Design

C57BL/6 mice (six-week-old, male) were obtained from Ziyuan Co., Ltd. (Hangzhou, China). All experimental procedures were conducted following the internationally accepted principles for the Care and Use of Laboratory Animals and approval from the Animal Research Committee of Zhejiang University (Protocol Number: 2023-1456). A managed environment with regulated humidity and temperature, featuring 12 h light/dark cycles, as well as specific pathogen-free (SPF) conditions, was applied to keep the mice. After a one-week acclimatization period, the mice were allocated randomly into three groups with 6 to 7 mice per group: the normal group (no APAP + 16% glycerol solution), the Control group (APAP + 16% glycerol solution), and the APAP + *C. minuta* group (APAP + *C. minuta* in a 16% glycerol solution). The mice were administered *C. minuta* (10^9^/mL) in a 16% glycerol solution or an equal amount of 16% glycerol solution via gavage for 14 days. On day 15, after undergoing a 16 h fasting period, the mice were intraperitoneally administered 300 mg/kg of APAP. Liver, colon tissues, and blood samples were collected for analysis 24 h after APAP administration.

To investigate the effects of phenylalanine, the mice were orally administered either 200 mg/kg of phenylalanine (Phe group), 200 mg/kg of phenylalanine combined with *C. minuta* (Phe + *C. minuta* group), or PBS (Con group) for a period of 14 days, and feces were collected for 16S rRNA sequencing. Subsequently, the mice received an intraperitoneal injection of 300 mg/kg of APAP after a 16 h fast. Liver, colon tissues, and blood samples were collected for analysis 24 h after APAP administration.

### 2.3. 16S rRNA Sequencing

DNA extraction was performed from the stool samples of mice gathered from mice across different experimental groups. The V3–V4 hypervariable regions were amplified utilizing the primers 341F (CCTACGGGNGGCWGCAG) and 806R (GGACTACHVGGGTATCTAAT). All purified amplicons underwent equimolar pooling and paired-end sequencing on an Illumina PE250 platform (Illumina, CA, USA). Subsequently, bioinformatics analysis of the raw data, including operational taxonomic units (OUT), community composition, α-diversity, β-diversity analysis, and function prediction, was conducted using various software tools, such as UPARSE (version 9.2.64), QIIME (version 1.9.1), and Muscle (version 3.8.31). Linear discriminant analysis (LDA) and effect size (LEfSe) analysis, statistical methods renowned for their discriminative power, were utilized to discern bacterial taxa exhibiting significant differences in abundance across the experimental groups. This discernment was based on stringent criteria, including a *p* value threshold of less than 0.05 and an LDA score exceeding 3, as determined through the robust nonparametric Kruskal–Wallis rank sum test. Additionally, Phylogenetic Investigation of Communities by Reconstruction of Unobserved States (PICRUSt) software (version 2.1.4) with Welch’s test analysis was utilized to forecast the pathways in functional genes of microbial communities across the groups. The Omicsmart online platform (http://www.omicsmart.com, accessed on 12 February 2024) served as a valuable tool for sequencing and conducting subsequent bioinformatics analysis.

### 2.4. Untargeted Metabolome Analysis

The serum from different groups was subjected to untargeted metabolome analysis by the UHPLC-Q Exactive HF-X system equipped with an HSS T3 column (Thermo Fisher, Waltham, MA, USA), following procedures outlined in previous research [14]. Briefly, serum samples underwent a series of extractions, followed by LC-MS/MS analysis, with data acquisition performed in either positive or negative ion mode (70–1050 *m*/*z*). After raw data underwent quality control, the sequencing information was compared and analyzed with existing databases such as The Human Metabolome Database (HMDB) and Metlin to determine metabolites. Majorbio cloud platform (https://cloud.majorbio.com, accessed on 3 January 2024) provided us with a lot of convenience for our subsequent bioinformatics analysis. Significant DMs were picked out in the light of the standard of variable importance in the projection (VIP) score > 1 from the orthogonal projections to latent structures discriminant analysis (OPLS-DA) model and a *p*-value < 0.05 based on the student’s *t*-test. Enrichment analyses of DMs were conducted utilizing the Kyoto Encyclopedia of Genes and Genomes (KEGG) (https://www.genome.jp/kegg/, accessed on 3 January 2024), and significance testing was conducted via Fisher’s exact test using scipy.stats (Python packages) (https://docs.scipy.org/doc/scipy/, accessed on 3 January 2024).

### 2.5. mRNA Sequence

Liver samples from various groups underwent high-throughput transcriptome sequencing, following previously described protocols [15]. Total RNA was extracted utilizing TRIzol reagent (Invitrogen, Carlsbad, CA, USA), followed by amplification, sequencing, and construction of RNA libraries on an Illumina sequencing platform (Gene Denovo Biotechnology, Co., Ltd., Guangzhou, China). Subsequently, after quality control and normalization, differentially expressed genes (DEGs) were screened with the help of the DESeq R package (version 1.32.0), applying criteria of log2|fold change| > 1 and *p*-value < 0.05. Additionally, the Short Time-series Expression Miner (STEM) software (version 1.3.13) was utilized to visualize and analyze the expression patterns of DEGs. Further analyses, including KEGG, GO, and GSEA, were conducted on Omicsmart (https://www.omicsmart.com, accessed on 3 March 2024.).

### 2.6. Biochemical Assay and ELISA

Serum ALT and AST levels were determined using an automatic blood biochemical analyzer (Beckman Coulter, Inc., Brea, CA, USA). To assess the levels of GSH, Malondialdehyde (MDA), superoxide dismutase (SOD), and inflammatory factors (IL6, IL10, TNF-α) in the liver, quantitative liver tissue sections were homogenized in phosphate-buffered saline (PBS) or the corresponding cracking solution, then centrifuged at 4000 rpm for 15 min. Various substances in the supernatant were measured using commercial kits, including the GSH and GSSG Assay Kit (Shanghai, China), as well as the Lipid Peroxidation MDA Assay Kit (Shanghai, China) purchased from Beyotime, the Total Superoxide Dismutase Assay Kit with WST-8 (Biosharp, Shanghai, China), and a Commercial ELISA kit (MLbio, Shanghai, China). These measurements were conducted following the protocols recommended by the respective manufacturers.

### 2.7. Histopathology, Immunohistochemistry and Terminal dUTP Nick-End Labeling (TUNEL) Stain

The liver and colon tissues were fixed in paraformaldehyde, embedded in paraffin, and cut into slices that are 7 μm thick. Following deparaffinization and rehydration, the tissue slides underwent staining with hematoxylin and eosin (H&E) to visualize cellular morphology and tissue architecture or incubated with primary monoclonal antibodies (anti-Occludin, anti-Claudin1, Proteintech, Wuhan, China) for immunohistochemistry. Subsequently, the slices were incubated with the goat anti-rabbit IgG (H + L) secondary antibody (Proteintech, Wuhan, China). Images were acquired using light microscopy (Axio Imager Z2, Carl Zeiss, Jena, Germany) after staining with a DAB Horseradish Peroxidase Color Development Kit (Beyotime, Shanghai, China). To detect DNA fragmentation using TUNEL staining, apoptotic cells exhibiting green fluorescence were discovered, guided by the manufacturer’s instructions for the One Step TUNEL Apoptosis Assay Kit (Beyotime, Shanghai, China).

### 2.8. Real-Time Quantitative PCR Analysis

The total RNA was extracted from liver tissue using the AFTSpin Tissue/Cell Fast RNA Extraction Kit for Animals (ABclonal, Wuhan, China) and was evaluated for both quality and quantity using NanoDrop Nucleic Acid Quantification (Thermo Fisher, USA). Subsequently, reverse transcription of RNA was performed using the ABScript III RT Master Mix (Abclonal, Wuhan, China). Real-time PCR was conducted using 2× Universal SYBR Green Fast qPCR Mix (Abclonal, Wuhan, China) on an Applied Biosystems Real-Time Quantitative PCR instrument, following the manufacturer’s recommended conditions. The primers for gene expression analysis were provided by Sangon Company (Shanghai, China), and the sequences were displayed in Appendix A. Cycle threshold (Ct) values were acquired after amplification, and the target gene expression was normalized to the internal control Glyceraldehyde-3-phosphate dehydrogenase (GAPDH). The comparative 2^−ΔΔCt^ method was applied to accurately compute the relative mRNA expression levels., with the results depicted as relative expression levels.

### 2.9. Cell Experiment In Vitro

The HepG2 cell line was maintained in our laboratory and cultured in DMEM with 10% FBS, 1% penicillin, and streptomycin. The HepG2 cell line was exposed to APAP at 20 mM for 24 h with or without *C. minuta* supplementation. Then, cellular viability was assessed utilizing the Cell Counting Kit-8 (CCK-8, GlpBio, Montclair, CA, USA) guided with the manufacturer’s protocol. Moreover, the level of lipid peroxidation was measured using BODIPY 581/591 C11 (GlpBio, CA, USA) by flow cytometry.

For the analysis of type of macrophages derived from the spleen, the spleen was chopped, ground, filtered through a 100 μm filter, and red blood cells were eliminated with Red Blood Cell Lysis Buffer (Beyotime, Shanghai, China). The remaining cells were washed with PBS, suspended in a complete culture medium, and seeded in a 6-well plate. After adhering to the wall for 6 h, the suspended cells were removed, and the remaining cells were stained with anti-CD86, anti-F4/80, anti-CD11b, and anti-CD206 (Biolegend, San Diego, CA, USA) for 15 min at room temperature. Finally, they were detected by cytoFLEX (Beckman Coulter).

### 2.10. Association Analysis of Microbiota, DEGs, DMs

Pearson’s correlation analysis was conducted to examine the relationship between the DEGs situated in profile 14 and the colon microbiota (Top10) at the genus level. Meanwhile, the relationships between microbes at genus level and metabolites were explored by Pearson’s correlation analysis. Co-expression networks were established using Cytoscape software (version 3.9.1) and OmicsNet (https://www.omicsnet.ca/, accessed on 10 April 2024), an effective tool to expand the connection among the distinct omics, was processed according to the previous research [16].

### 2.11. Statistical Analysis

The experimental data underwent analysis via Graphpad 8.0. Unpaired two-tailed Student’s *t*-tests were employed for comparing two distinct groups, whereas for comparisons beyond two groups, analysis of variance (ANOVA) was performed. Significance was indicated by *p* < 0.05 and the results were expressed as mean ± standard deviation.

## 3. Results

### 3.1. C. minuta Supplementation Mitigated AILI

Assessing *C. minuta*’s impact on APAP toxicity, the supplement (10^9^ CFU/mL) was orally administered to mice over a span of 14 days preceding the administration of APAP (300 mg/kg) (Figure 1A). Pivotal biomarkers of hepatic injury, ALT and AST [1], were significantly reduced by *C. minuta*, indicating its efficacy in lowering the elevated serum levels caused by APAP (Figure 1B,C). Furthermore, H&E staining corroborated the supplement’s role in alleviating hepatic damage from APAP, notably through diminished centrilobular necrosis (Figure 1D,E). TUNEL assays, aimed at detecting nuclear DNA fragmentation indicative of hepatocellular apoptosis triggered by APAP hepatotoxicity [17], indicated a reduction in TUNEL-positive hepatocytes in the APAP + *C. minuta* group compared with the APAP + PBS group, thereby suggesting a mitigated apoptotic response within the liver (Figure 1F). 

### 3.2. C. minuta Supplementation Relieved Oxidative Stress and Inflammatory Response

GSH plays a pivotal role in neutralizing NAPQI’s toxic impact and mitigating oxidative stress. Our observations unveiled a noteworthy decline in hepatic GSH levels and GSH/GSSG ratio in mice exposed to APAP, while treatment with *C. minuta* elicited a conspicuous elevation in GSH levels and the GSH/GSSG ratio (Figure 2A). Beyond GSH, *C. minuta*’s beneficial effect in mice exposed to APAP extended to enhancing the activity of essential antioxidant enzymes like SOD (Figure 2B) and lowering MDA (Figure 2C), an indicator of lipid peroxidation. Additionally, lipid peroxidation assessments via flow cytometry in HepG2 cells showed an uptick in FITC fluorescence following APAP exposure, suggesting increased lipid peroxidation, which was notably decreased by *C. minuta* supplementation (Figure 2D). Then, we detected the inflammation levels in liver tissue at both the protein and mRNA levels. The pro-inflammatory factors (Il1β, Il6, Tnfα) were upregulated in APAP-exposed mice but alleviated with *C. minuta* treatment (Figure 3A,B). Macrophages, as innate immune cells, play a pivotal role in the progression of liver damage [18,19]. The imbalance of the M1/M2 ratio in splenic-derived macrophages was evidenced by a significant increase in M1 type in APAP-exposed mice, while in the APAP + *C. minuta* group, the proportion of M1 decreased (Figure 3C). In the APAP group, the number of M2 was more than in the APAP + *C. minuta* group, possibly due to compensatory increases in response to excessive pro-inflammatory factors. Considering the ectopic intestinal bacteria-induced inflammation due to mucosal barrier damage, we assessed the barrier function in the colon. We only observed increased immune cell infiltration with APAP administration (Appendix A), but no significant change in Claudin-1 and Occludin expression (Appendix A), which are critical markers of intestinal permeability.

### 3.3. C. minuta Regulated the Phenylalanine Metabolism

Considering *C. minuta*’s known role in regulating metabolic-related diseases such as obesity and T2D, we detected the metabolites altered by *C. minuta* supplementation in mouse serum using non-targeted metabolomics. Following data quality control, 948 primary metabolite components were identified. The different metabolic profiles between groups treated with or without *C. minuta* was observed through an OPLS-DA score plot (Figure 4A). Using VIP > 1 and a *p* value < 0.05 threshold, *C. minuta* supplementation increased 42 metabolites and decreased 58 metabolites in positive ion patterns, while increased 15 products and declined 67 products in negative ion patterns (Figure 4B). Additionally, the unsupervised hierarchical clustering heatmap illustrated the top 50 DMs (Figure 4C). Subsequently, Spearman’s correlation analysis was performed between the top 50 DMs and ALT and AST, revealing that lipid metabolites (e.g., LysoPE(18:0/0:0) and PE(18:0/0:0)) and phenylalanine metabolites (e.g., Phenylacetic acid, L-phenylalanine) displayed positive correlations with liver injury. KEGG enrichment analysis was conducted based on the DMs, revealing various pathways influenced by *C. minuta* (showed top 20, Figure 4D), such as FoxO signaling pathway, phenylalanine metabolism. FoxO is reported to regulate cell metabolism, proliferation, differentiation, apoptosis, oxidative stress, DNA damage, and repair [20,21]. Elevated levels of phenylalanine and its metabolites were found to be harmful for Alzheimer’s disease progression [22], and alcohol-related liver disease [23]. Overall, these findings suggested that *C. minuta’*s protective effects may contribute to modulating oxidation, inflammation, regeneration pathways, and phenylalanine metabolism.

### 3.4. Phenylalanine Supplementation Aggravated AILI

High phenylalanine levels have been reported to be associated with an increased risk of various diseases, such as cardiovascular disease [24] and alcoholic fatty liver disease [23]. However, the impact of phenylalanine on liver damage and whether *C. minuta* mitigates AILI by modulating phenylalanine metabolism remain completely unknown. Therefore, mice were administered phenylalanine for 14 days via gavage (Figure 5A), and their food intake and weight were meticulously monitored daily. Remarkably, the mice tolerated phenylalanine treatment well, with no significant change observed in weight loss or daily dietary intake (Figure 5B,C). After being exposed to APAP, mice receiving daily doses of phenylalanine exhibited more severe hepatocyte damage, as evidenced by elevated serum ALT and AST levels (Figure 5D), more pronounced cell ballooning, and a greater area of necrosis (Figure 5E,F), along with more TUNEL-positive cells than mice that received only APAP administration (Figure 5G). In line with our hypothesis, under the dual impact of phenylalanine and APAP, *C. minuta* still effectively alleviated liver injury but could not reduce the damage to the same level as that in the Con group. This indicated that *C. minuta* indeed partially alleviated damage by regulating phenylalanine and that the damage caused by phenylalanine exceeded the scope of *C. minuta* regulation.

### 3.5. C. minuta Supplementation Improved High Oxidative Stress Levels Exacerbated by Phenylalanine in AILI

To investigate whether elevated levels of phenylalanine exacerbate oxidative stress in the body, we observed a distinct reduction in GSH and GSH/GSSG levels with phenylalanine supplementation in an AILI model, while *C. minuta* restored GSH losses. However, no statistically significant difference was observed in GSSG levels (Figure 6A). SOD activity decreased and MDA levels increased in mice exposed to both APAP and phenylalanine compared to those receiving APAP alone, and *C. minuta* partially restored the antioxidative condition caused by phenylalanine overdose (Figure 6B,C). Flow cytometry also revealed that the combination of APAP and phenylalanine resulted in the highest level of lipid peroxidation, which was ameliorated by *C. minuta*. Interestingly, high phenylalanine treatment alone led to a moderate increase in intracellular lipid peroxidation levels. As for the pro-inflammatory level in liver and colonic barrier function, we only observed a slight increase in inflammation level and more immunocytes infiltration when phenylalanine was administered in the AILI model (Appendix A).

### 3.6. The MAPK Signaling Pathway Was the Likely Focus of Phenylalanine and C. minuta

RNA sequencing is a common tool used to explore underlying mechanisms at the transcriptional level. Therefore, we conducted high-throughput screening of liver tissue. Correlation analyses revealed that samples in each group had a correlation coefficient > 0.9 for Pearson’s analysis, indicating high experimental repeatability (Figure 7A). Additionally, the PCA plot demonstrated that the three groups were clearly distinguishable, suggesting different expression profiles among the groups (Figure 7B). By applying strict screening criteria with a *p*-value < 0.05 and |log2FC| > 1, we identified 1410 DEGs (1061 upregulated, 349 downregulated) when comparing groups Control and Phe. Similarly, 164 DEGs (109 upregulated, 55 downregulated) were identified when comparing groups Phe and Phe + *C. minuta* (Figure 7C). Using STRING, Cytoscape and MCODE tools, hub genes within the large network were identified to explore phenylalanine’s toxicity to AILI, focusing on cytochrome P450 metabolism, sphingolipid metabolism and other signaling pathways. Consistent with KEGG analysis, higher phenylalanine levels resulted in various metabolic disorders, such as drug, lipid, glutathione, and multiple signal pathway abnormalities, including MAPK and PI3K-AKT signaling pathways (Figure 7E). To address the issue of how *C. minuta* mitigated hepatotoxicity exacerbated by phenylalanine, we focused on genes with changing trends across the three groups. The heatmap displays common DEGs in pairwise comparisons (Figure 7F). Furthermore, the DEGs underwent STEM analysis to identify genes potentially influenced by phenylalanine and *C. minuta*, revealing the presence of four distinct module. Attention was paid to genes in Profile 14, which were upregulated in group Phe but alleviated by *C. minuta* (Figure 7G and Appendix A). GO and KEGG analysis revealed modulation of oxidative stress and the MAPK signaling pathway in Profile 14 (Figure 7H,I).

### 3.7. C. minuta Modulated the Gut Microbiota Disturbed by Phenylalanine

The microbiota exerts influence on distal organ function through the action of their metabolites or the immune system. Dysbiosis, triggered by environmental factors like alcohol consumption, may result in intestinal barrier dysfunction, translocation of microbial components to the liver, and the onset or exacerbation of liver disease [7]. To elucidate the roles of gut microbiota in liver injury, we employed 16S rDNA gene sequencing technology to analyze variations in their abundance. Initially, we assessed alpha diversity using Shannon, Simpson, Chao, and Sob indices, revealing no statistical differences among the three groups (Appendix A). However, the Venn plot displayed the lowest microbial diversity at the genus level in the Phe group (Figure 8A), indicating that phenylalanine supplementation reduced bacterial diversity. Principal coordinates analysis (PCoA) plots of weighted UniFrac distances and β diversity statistical testing indicated remarkable differences in bacterial composition. Notably, microbial community compositions were significantly reduced with phenylalanine supplementation, while administration of *C. minuta* increased the diversity of microbial community structure (Figure 8B,C). Upon taxonomic composition analysis, a dramatic rise in the relative abundance of Firmicutes at phylum level, particularly Lactobacillus at genus level, was noticed with phenylalanine treatment compared to the other groups. Conversely, *Lactococcus*, *Acinetobacter*, and *Akkermansia* were reduced in the Phe group but restored in the Phe + *C. minuta* group (Figure 8D,E). Furthermore, we conducted LEfSe analysis to explore notable microbial taxa among different groups. The abundance of *Lactobacillus murinus* (*L. murinus*) significantly increased in phenylalanine-fed mice compared to the Control and Phe + *C. minuta* groups, while Lactococcus dominated in the Control group (Figure 8F). *C. minuta* supplementation disrupted the absolute dominance of L. murinus in the phenylalanine group, with several probiotics showing increased abundance, such as *Clostridium sensu stricto* 1 [25], *Bacteroides* [26], *Oscillospira* [27] which were reported beneficial for liver disease. Subsequently, PICRUSt2 analysis was performed to forecast the biological functions of the microbiota (Figure 8G). Atrazine degradation was upregulated, while steroid hormone biosynthesis was inhibited with phenylalanine supplementation. Conversely, feeding with *C. minuta* combined suppressed the phosphotransferase system and atrazine degradation, while enhancing apoptosis, the NOD-like receptor signaling pathway, and protein digestion and absorption (Figure 8H).

### 3.8. Integrated Analysis

To explore the connection between fecal microbiota and liver transcriptome, we conducted Pearson’s correlation analysis between main bacteria and genes in profile 14. *Lactobacillus* was positive with Gm20521, Car6, Mucl2, Igf2, Mmrn1, and Gm49948, and was negative with Tuba8 and Rec8. *Lactococcus* was positively correlated with Upp2, Dntt, and Lgr5 (Figure 9A). Subsequently, we explored the correlation between gut microbiota and the DMs in phenylalanine metabolism with Pearson correlational analysis. We found that *Lactobacillus* showed a positive correlation with most metabolites of phenylalanine, while *Parasutterella*, *Oscillibacter*, and *Lachnoclostridium* was negative with most metabolites of phenylalanine (Figure 9B). Finally, we used the web tool Omicsnet to link the three omics and discover the mechanism of *C. minuta* through the gut-liver axis. From the results, the modules were explored in the large net via InfoMap Algorithm, and four modules were identified: drug metabolism-cytochrome P450 and various animo metabolisms were regulated by *C. minuta*, including the phenylalanine (Figure 9C).

## 4. Discussion

DILI poses a significant challenge in clinical treatment due to drug abuse. Certain patients with susceptibility heightened even at therapeutic doses, particularly those exposed to risk factors such as alcohol, may suffer severe liver injury, or even liver failure [28]. Despite residing in the intestine, the microbiota serve as a vital link between extra-intestinal sites and the gut, maintaining bodily homeostasis through metabolic and immunological mechanisms. Notably, because the liver is the first organ to encounter substances from the intestinal mucosa, dysbiosis is considered a causative or predisposing factor in liver diseases [7,29]. Recent investigations have delved into the role of gut microbiota in AILI, and probiotic interventions have demonstrated promising therapeutic effects through various pathways. Gut microbiota dysbiosis exacerbated hepatotoxicity in mice treated with APAP by promoting hepatocyte ferroptosis mediated by Fdps [30]. *Lactobacillus rhamnosus GG* alleviated APAP hepatotoxicity, partly through nuclear factor erythroid 2-related factor 2 (Nrf2), an essential controller of antioxidant reactions [31]. Similarly, *Bifidobacterium longum R0175* mitigated hepatotoxicity via attenuating inflammation, reducing oxidative stress though Nrf2 pathway activation, inhibiting hepatocyte death, and ameliorating APAP-induced microbiome dysbiosis [32]. Encouragingly, our research has demonstrated that supplementation with *C. minuta* enhances the liver’s resilience against APAP toxicity by restoring the balance of gut microbiota and increasing diversity. Previous studies have underscored the critical role of *C. minuta* in regulating energy balance and metabolic homeostasis [13,33,34]. Besides restoring microbial balance, *C. minuta* carries potent bile salt hydrolase (BSH) activity, which preferentially deconjugates glycine-conjugated BAs. This activity helps *C. minuta* resist bile acids, facilitating successful colonization of the GI tract environment [35]. Additionally, it enables the reabsorption of unconjugated BAs by passive diffusion and allows further metabolism by bacteria to produce secondary BAs (e.g., deoxycholic and lithocholic acid) [36]. One study showed that BAs reduced liver injury during the initiation of AILI [37]. However, other studies have indicated that bile acids enhance TNF-α release and potentiate APAP toxicity. Therefore, preventing bile acid cycling may represent a therapeutic option after APAP intoxication [38,39]. Otherwise, NAPQI is produced as the oxidative metabolite of APAP through the cytochrome P450 system, disrupting mitochondrial energy metabolism and escalating oxidative stress within the body [28]. This metabolic perturbation prompts a focus on the regulatory effects of probiotics on metabolism. Non-targeted metabolomics analyses have revealed alterations in the phenylalanine metabolic pathway, with phenylalanine and its metabolite levels showing significant negative correlations with ALT and AST levels.

Multiple factors could contribute to elevated phenylalanine levels in the body, including the consumption of aspartame [40], Phenylketonuria [41], and liver dysfunction [42]. Furthermore, phenylalanine has been reported to correlate with the incidence and severity of alcoholic hepatitis [23]. L-phenylalanine effectively suppresses the activity of intestinal alkaline phosphatase (IAP). Known for its capacity to diminish endotoxemia and subsequent inflammation while also maintaining intestinal barrier integrity [23,43], IAP has emerged as a promising therapeutic element in addressing conditions such as alcohol-induced liver steatosis and inflammation, as well as in ameliorating bile duct ligation and carbon tetrachloride-induced liver fibrosis [44]. Additionally, phenylalanine has an inhibitory effect on indole-3-propionic acid (IPA) in the liver and intestine, disrupting gut microbiota structure. We elucidated the impact of phenylalanine on liver health and whether modulation of this metabolism constitutes a mechanism through which *C. minuta* confers therapeutic benefits for liver damage. Our findings revealed that prior to APAP exposure, phenylalanine intake alone did not influence body weight and food intake but uniquely altered the composition of the intestinal microbiota, diminished bacterial diversity, and fostered overgrowth of *L. murinus*. This phenomenon may be attributed to phenylalanine provision, which prompts a compensatory increase in *L. murinus*—a primary phenylalanine-metabolizing bacterium known to impair gut metabolic function and contribute to alopecia development [45]. Supplementation with *C. minuta* disrupted the dominance of *L. murinus*, enhanced microbial diversity, and increased the abundance of several beneficial microbes. Restoration of gut microbiota diversity may constitute a crucial aspect of *C. minuta*’s functionality. Our study demonstrated a significant enhancement in atrazine metabolism by bacteria when phenylalanine was provided, which may augment CYP450 activity since atrazine is metabolized by CYP450 [46]. This enhancement potentially enables more APAP to be metabolized into NAPQI through the CYP450 system, exacerbating liver damage. Additionally, bacterial dehydration of atrazine to cyanuric acid further compounds its harmful effects on the body [47].

To explore pathways through which phenylalanine exacerbates AILI and how *C. minuta* regulates phenylalanine metabolism, mRNA sequencing was performed. Phenylalanine has been found to interfere with multiple metabolic pathways, including cytochrome P450 metabolism, steroid hormone metabolism, retinol metabolism, glutathione metabolism, primary bile acid metabolism, sphingolipid metabolism, and nitrogen metabolism. Additionally, it affects signaling pathways such as PI3K-AKT, MAPK, and the integrin pathway. Using STEM analysis, we explored potential genes and pathways that *C. minuta* may act upon, revealing its ability to respond to oxidative stress, regulate lipid metabolism, and modulate the MAPK signaling pathway. Considering the pathogenesis of AILI involves ROS generation in mitochondria amplified through c-Jun *N*-terminal kinase (JNK) [48], our research demonstrated that elevated phenylalanine levels increased oxidative stress and activated the MAPK pathway, while *C. minuta* supplementation reduced oxidative stress and inflammation. This suggests that the MAPK signaling pathway plays a critical role in exacerbating phenylalanine-induced hepatocyte injury or alleviating it with *C. minuta*. In summary, the consumption of phenylalanine gradually alters our gut microbiota and certain genes, making them more sensitive to risk factors such as alcohol and drugs, thereby exacerbating liver damage. Incorporating probiotics into daily life may prevent dysbiosis and broaden the applications of probiotics in managing liver health.

High levels of phenylalanine itself have been shown to increase inflammatory factors TNF-α and IL-6 [23], leading to excessive necroptosis in macrophages and the formation of abundant ROS [49]. Additionally, phenylalanine’s metabolite phenylacetic acid (PAA) is harmful to the body as it induces hepatic steatosis and increases the utilization of branched-chain amino acids [50]. In healthy subjects, elevated levels of fecal PAA coincide with an increased abundance of the *Clostridium XIVa cluster* and decreased presence of *Lactobacillus*. Furthermore, heightened fecal phenylpropionate levels have been linked to elevated serum concentrations of transforming growth factor (TGF)-β, IL-17, IL-8, MDA, and *C*-reactive protein (CRP) [51]. High levels of inflammation promote the progression of liver diseases [52]. Following APAP administration, the phenylalanine group exhibited heightened inflammation and liver injury compared to those administered APAP alone. However, *C. minuta* supplementation attenuated inflammation and oxidative stress. In addition, amino acid metabolism is highly complex in biological systems. Phenylalanine can be converted to tryptophan [53] and interacts with branched-chain amino acids (BCAAs, leucine, isoleucine, and valine), glutamate, and alanine. Tryptophan is converted into various neurotransmitters, such as serotonin, which has shown a protective effect against AILI [54]. While BCAAs, glutamate, and alanine have been linked to metabolic dysfunction-associated fatty liver disease (MAFLD) and obesity [55,56], their specific roles in AILI remain understudied. However, GSH, composed of glutamic acid, cysteine, and glycine, is an antioxidant molecule that plays a protective role in liver injury [57]. Unfortunately, our investigation did not quantify alterations in these metabolites or other interacting amino acids within the serum and liver. Recently, a special type of DILI characterized by predominant GGT elevation leading to drug withdrawal and/or chronic elevation of liver parameters, particularly GGT, with liver enzyme elevation below conventional thresholds, has been reported. However, this aspect was overlooked in this study as GGT levels were not tested. Further investigation is needed to determine whether phenylalanine’s metabolites or other amino acids contribute to hepatic injury during AILI, and to assess changes in GGT levels in the experiments. Furthermore, further exploration is needed to apply *C. minuta* in clinical practice. It is crucial to recognize differences between species that may affect probiotic efficacy and metabolism, and to ensure that the probiotic strains and dosages used in animal studies are relevant and safe for human use. Based on these considerations, rigorous, well-controlled human clinical trials are essential to confirm safety and efficacy, followed by standardized industrial production. 

## 5. Conclusions 

Although more detailed experiments are needed, the impact of *C. minuta* on the microbiome and various metabolic changes in the host is significant. Notably, *C. minuta* reduces inflammation and oxidative stress by regulating phenylalanine metabolism. Elevated levels of phenylalanine exacerbated AILI by inducing dysbiosis, increasing ROS, and activating the MAPK signaling pathway, all of which were mitigated by *C. minuta* supplementation (Figure 10). Our study not only demonstrates the potential of *C. minuta* in treating AILI but also suggests that phenylalanine sensitizes individuals to AILI—a sensitivity that can be alleviated by *C. minuta*—providing a potential treatment method for AILI.

## Figures and Tables

**Figure 1 nutrients-16-02314-f001:**
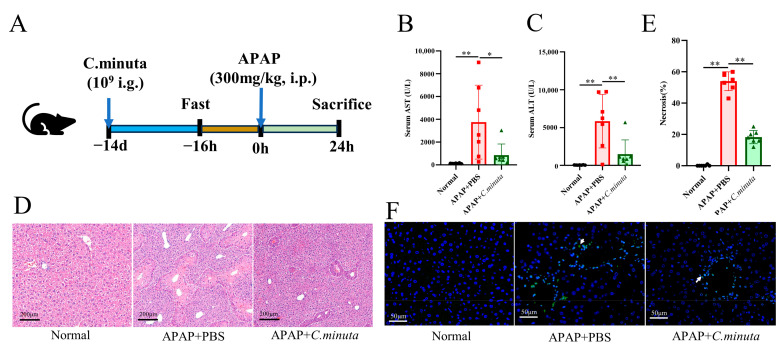
*C. minuta* supplementation mitigates AILI in mice (n = 7). (**A**) Schematic diagram of the animal experiment design. (**B**,**C**) Serum levels of ALT and AST. (**D**,**E**) Representative images of H&E staining in mice livers (scale bar = 50 μm) and quantification of the area of necrosis. (**F**) Representative liver DAPI/TUNEL immunofluorescence staining (scale bar = 50 μm; white arrow indicates TUNEL-positive cells). * *p* < 0.05, ** *p* < 0.01.

**Figure 2 nutrients-16-02314-f002:**
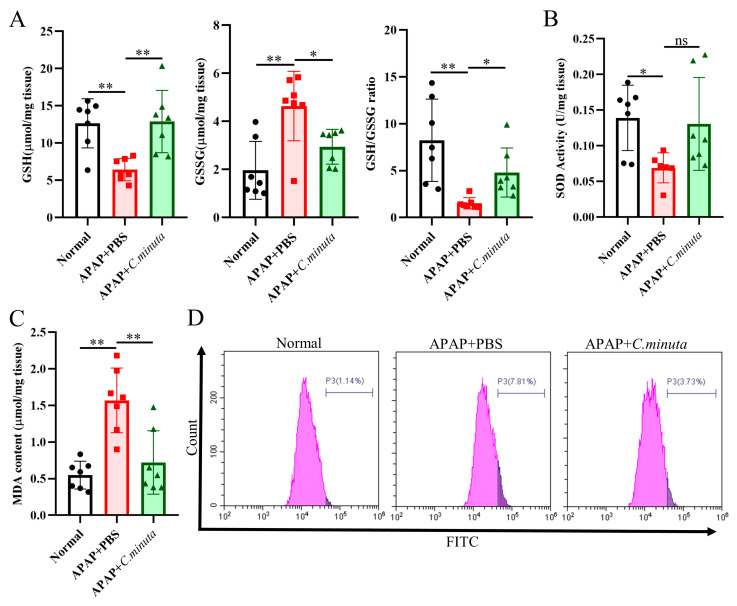
*C. minuta* supplementation relieved oxidative stress in mice (n = 7). (**A**) GSH level, GSSG level, and GSH/GSSG ratio were measured in liver tissue. (**B**) SOD activity was assessed in liver tissue. (**C**) MDA content was quantified in liver tissue. (**D**) MDA content in HepG2 cells was evaluated using BODIPY 581/591 C11 for flow cytometry detection. * *p* < 0.05, ** *p* < 0.01, ns: no significance.

**Figure 3 nutrients-16-02314-f003:**
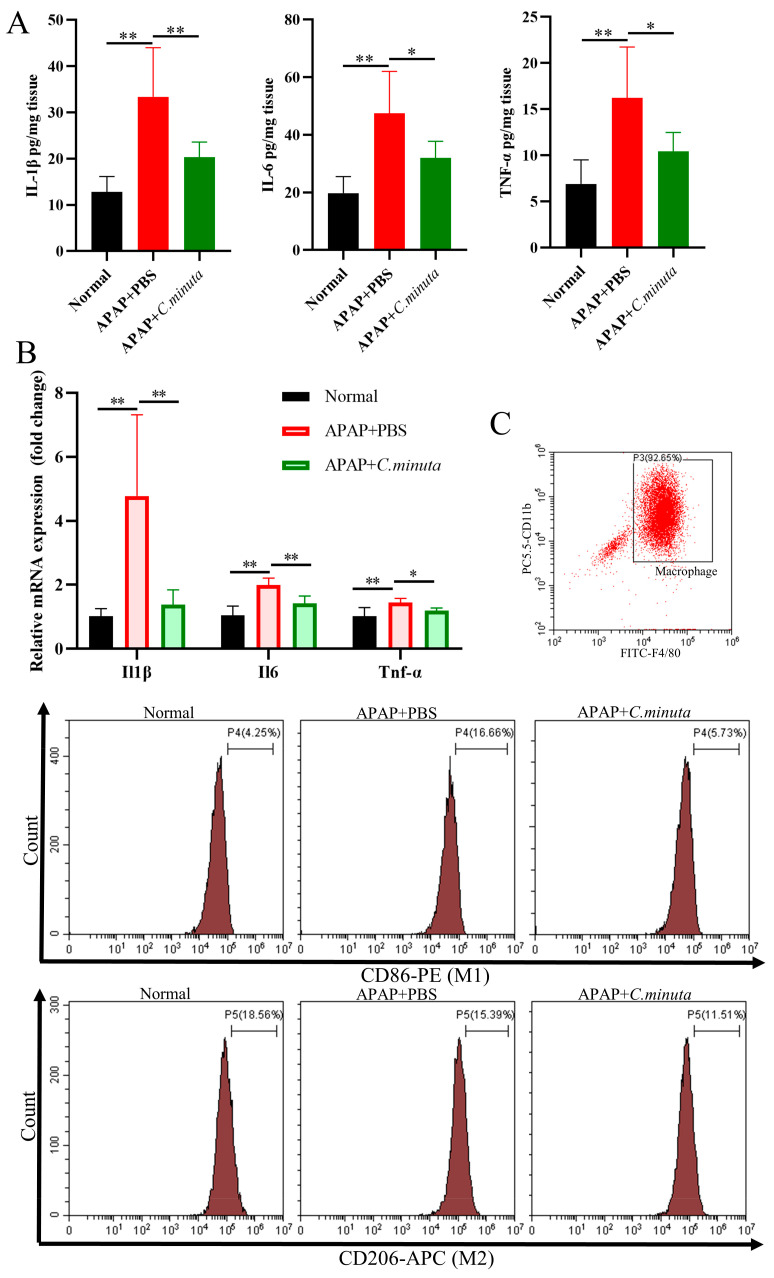
*C. minuta* supplementation relieved inflammatory response. (**A**) Protein levels of IL1β, IL6, TNF-α in mice liver were measured by ELISA. (**B**) mRNA levels of Il1β, Il6, Tnfα in mice liver were detected by qRT-PCR. (**C**) The proportion of M1 and M2 macrophages in splenic-derived macrophages. * *p* < 0.05, ** *p* < 0.01.

**Figure 4 nutrients-16-02314-f004:**
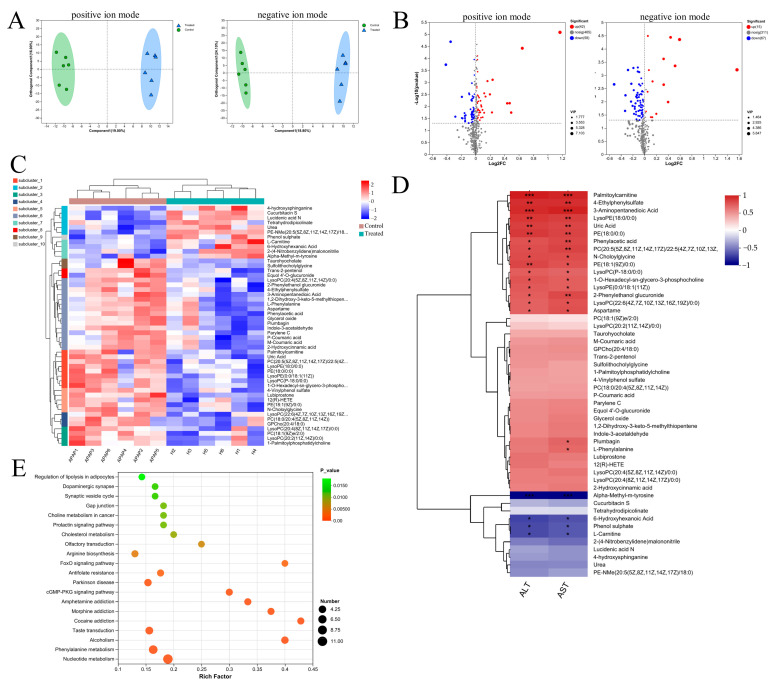
Metabolic profiling in serum from APAP-exposed mice treated with or without *C. minuta*. (**A**) orthogonal projections to latent structures-discriminant analysis (OPLS-DA) score plots in positive and negative ionization modes. (**B**) Volcano plot for different metabolites in positive and negative ion modes. (**C**) Unsupervised hierarchical clustering heatmap of the top 50 different metabolites. (**D**) Spearman’s correlation analysis between the top 50 DMs and ALT and AST. (**E**) Top 20 pathways analyzed by KEGG. Control in the figure represents the APAP + PBS group; Treated represents the APAP + *C. minuta*. * *p* < 0.05, ** *p* < 0.01, *** *p* < 0.001.

**Figure 5 nutrients-16-02314-f005:**
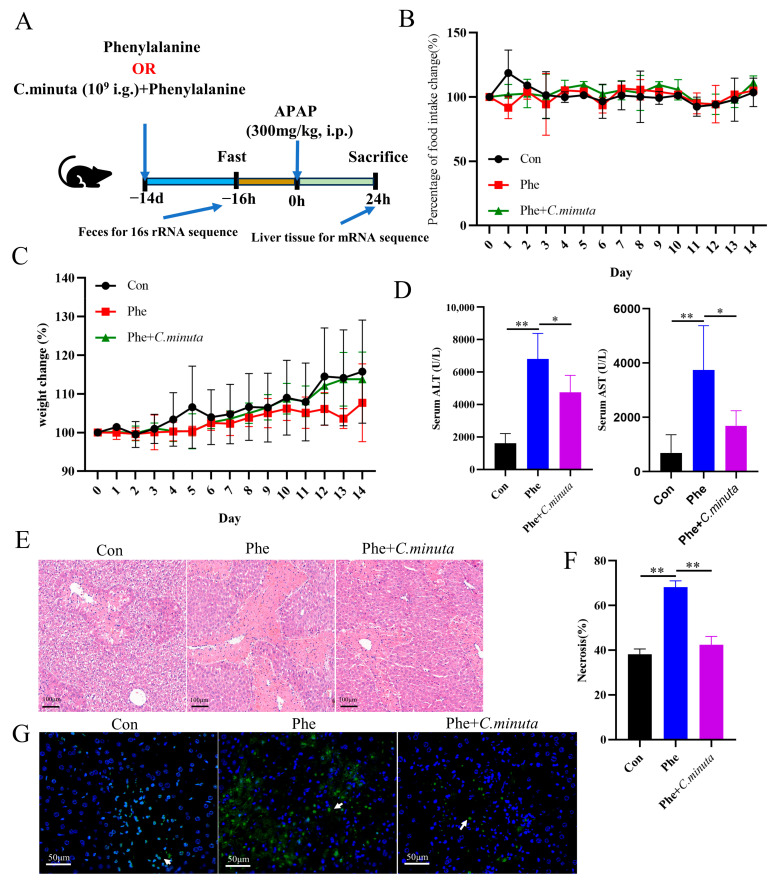
Phenylalanine supplementation aggravated AILI (n = 5). (**A**) Schematic diagram of the animal experiment design. (**B**) Daily food intake change in mice. (**C**) Daily weight change. (**D**) Serum levels of ALT and AST. (**E**,**F**) Representative images of H&E staining in mice livers (scale bar = 100 μm) and quantification of the area of necrosis. (**G**) Representative liver DAPI/TUNEL immunofluorescence staining (scale bar = 50 μm; white arrow indicates TUNEL-positive cells). * *p* < 0.05, ** *p* < 0.01.

**Figure 6 nutrients-16-02314-f006:**
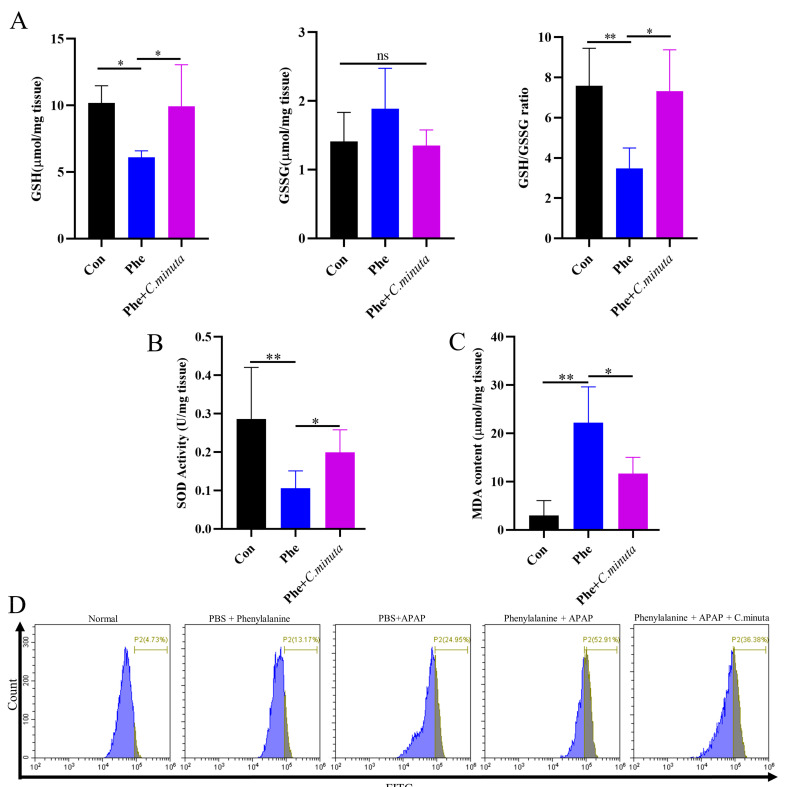
*C. minuta* improves high oxidative stress levels exacerbated by phenylalanine in AILI (n = 5). (**A**) GSH level, GSSG level, and GSH/GSSG ratio were measured in liver tissue. (**B**) SOD activity was assessed in liver tissue. (**C**) MDA content was quantified in liver tissue. (**D**) MDA content in HepG2 cells was evaluated using BODIPY 581/591 C11 for flow cytometry detection. * *p* < 0.05, ** *p* < 0.01.

**Figure 7 nutrients-16-02314-f007:**
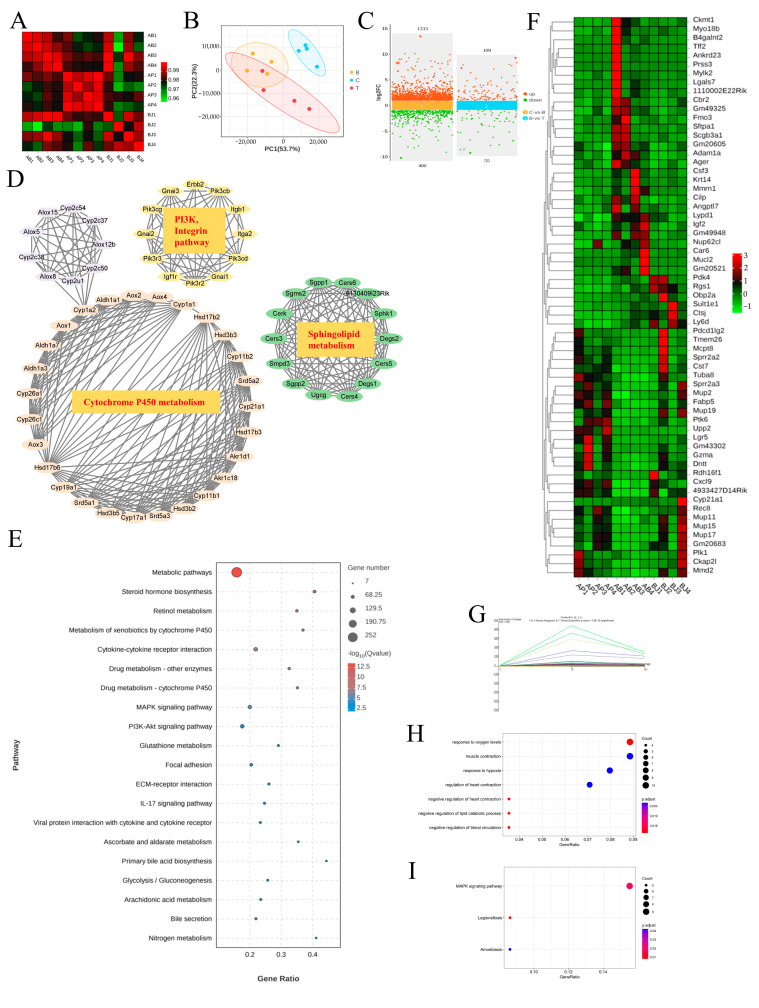
The MAPK signaling pathway was the potential target of phenylalanine and *C. minuta*. (**A**) Sample correlation test. (**B**) Principal component analysis (PCA) of individual samples for the normalized mRNA data in liver tissue. (**C**) Plot of differentially expressed genes. (**D**) Hub genes analyzed by Cytoscape and Mode among DEGs compared with groups B and T. (**E**) KEGG analysis of DEGs compared with groups B and T. (**F**) The common DEGs in pairwise comparisons. (**G**) Trend graph of all genes under profile 14, where the thick green curves indicate the expression trend across all genes in a module. (**H**) GO analysis of profile 14. (**I**) KEGG analysis of profile 14. Group B represents Phe group; Group T represents Phe + *C. minuta* group.

**Figure 8 nutrients-16-02314-f008:**
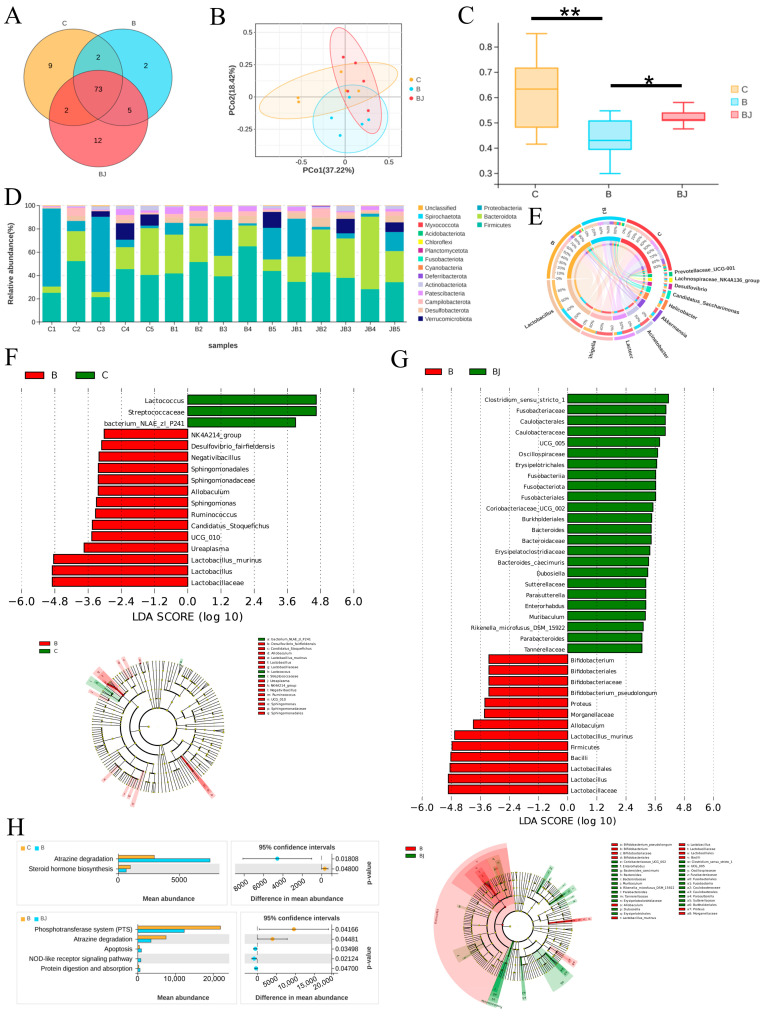
*C. minuta* modulates the gut microbiota disturbed by phenylalanine (n = 5). (**A**) The Venn plot of microbiota at the genus level in different groups. (**B**) PCoA analysis showed that the microbiota were clustered differently among the three groups. (**C**) β diversity testing by Welch’s *t*-test analysis. (**D**) The relative abundance at the phylum level in each sample. (**E**) The relative abundance at the genus level in each sample. (**F**,**G**) LEfSe (linear discriminant analysis effect size) dendrogram was used to analyze the classification tree of sample species and find the marker species with significant differences in different groups; LDA = 3. (**H**) PICRUSt2 analysis predicted the microbiota’s function in different groups. C represents the Control group, B represents Phe group, BJ represented Phe + *C. minuta* group. * *p* < 0.05, ** *p* < 0.01.

**Figure 9 nutrients-16-02314-f009:**
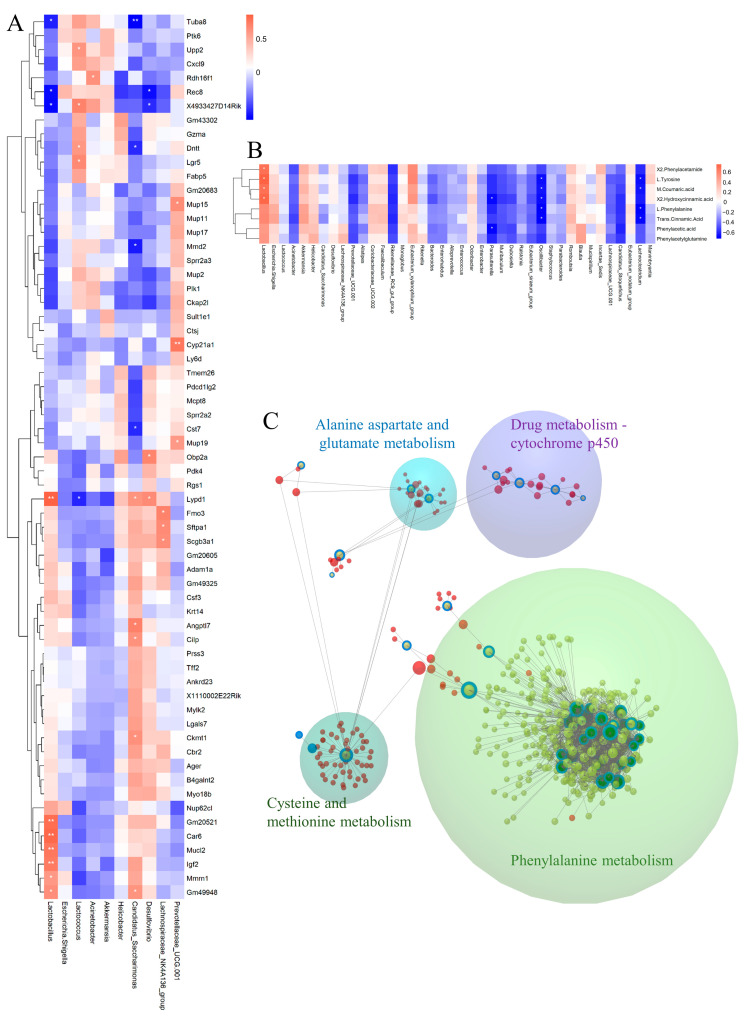
Multiomics integrated analysis. (**A**) The Pearson correlation analysis between microbes and DEGs. (**B**) The Pearson correlation analysis between microbes and DMs within phenylalanine metabolism. (**C**) The analysis of the relationship between three DEGs, DMs, and microbiota by Omicsnet. (Green dots represent microbes, red dots represent mRNA/protein, yellow dots represent metabolites. Dots with a blue ring represent substance data input from our original source, while dots without a blue ring represent substance data expanded by the software itself.) * *p* < 0.05, ** *p* < 0.01.

**Figure 10 nutrients-16-02314-f010:**
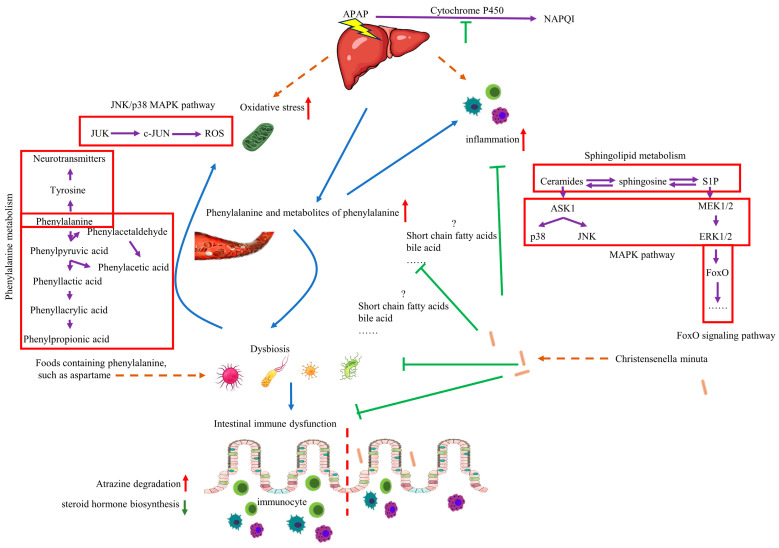
The mechanism of function of *C. minuta* and phenylalanine on AILD. Administration of phenylalanine leads to disruption of gut microbiota, making the liver more susceptible to drug damage, increasing inflammation and oxidative stress levels. *C. minuta* regulates phenylalanine metabolism, restores gut microbiota diversity, regulates MAPK pathway, reduces inflammation and oxidative stress levels, and alleviates liver damage finally.

## Data Availability

Data are contained within the article or Appendix A.

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
