# Peer review of "Christensenella minuta Alleviates Acetaminophen-Induced Hepatotoxicity by Regulating Phenylalanine Metabolism"

_nutrients, 2024, doi:10.3390/nu16142314_

Round 1

Reviewer 1 Report

Comments and Suggestions for Authors

This study has examined the efficacy of the intestinal microbe, Christensenella minuta, for alleviating liver injury induced by acetaminophen (APAP). It is suggested that the APAP induced hepatotoxicity is caused by accumulations of phenylalanine and its metabolites. The treatment of C. minuta could prevent the accumulations of them, and become an effective treatment for liver injury. Regarding the mechanism, the authors have suggested that the efficacy of the microbe supplement could be attributed to its anti-inflammatory and anti-oxidative effects. Comments and questions are written as follows.

1. Christensenella minuta is an intestinal microbe, too. Is the gut indigenous C. minuta influenced somehow by its treatment?

2. In section 3.3, the efficacy of C. minuta itself should be described more.

3. In section 3.4, could the metabolites of phenylalanine be accumulated by phenylalanine treatment, too?

4. The authors have discussed the involvement of aspartame, as a phenylalanine compound, in liver injury. Could phenylalanine metabolites be actually accumulated by aspartame intake?

5. Check carefully axes titles. For example, in Figure 2A, is “GSSG” right?

Reviewer 2 Report

Comments and Suggestions for Authors

Other authors identified and cloned a protein from C. minuta’s genome that carries a potent bile salt hydrolase (BSH) activity, which preferentially deconjugates glycine-conjugated bile acids...as evident in....Microorganisms 20219(6), 1252; https://doi.org/10.3390/microorganisms9061252

Could this mechanism be implicated in the therapeutical approach to this model of hepatoxicity ?

Please, discuss this aspect.

Authors should emphasize that....DILI is rare but can be serious and is "largely unpredictable"....as evident in.. Drug-induced liver injury, dosage, and drug disposition: is idiosyncrasy really unpredictable? Clin Gastroenterol Hepatol. 2014 Sep;12(9):1556-61. doi: 10.1016/j.cgh.2014.02.011. Epub 2014 Feb 12. PMID: 24530601.

Authors presented a case series of patients with liver enzyme elevation below the conventional thresholds who developed DILI with a predominant GGT elevation leading to drug withdrawal and/or chronic elevation of liver parameters, in particular of GGT. Thus, they propose that DILI should be considered in particular in cases with marked increase of GGT even if conventional DILI threshold levels are not reached, resulting in discontinuation of the causative drug and/or close monitoring of the patients...as evident in..Marked Increase of Gamma-Glutamyltransferase as an Indicator of Drug-Induced Liver Injury in Patients without Conventional Diagnostic Criteria of Acute Liver Injury. Visc Med. 2022 Jun;38(3):223-228. doi: 10.1159/000519752. Epub 2021 Nov 3. PMID: 35814980; PMCID: PMC9209957.

Have authors data on levels of GGT and can they show them?

If not available, this is a limitation to study.

Comments on the Quality of English Language

Acceptable!

Reviewer 3 Report

Comments and Suggestions for Authors

Manuscript title: Christensenella minuta alleviates acetaminophen induced 2

hepatotoxicity by regulating the phenylalanine metabolism

This manuscript addresses the ameliorative effects of supplementing of C. minuta on the acetaminophen (APAP) induced liver injury (AILI). Overall, the manuscript has clearly presented the reasonable results and the methods used in the study are appropriate to the aims of the study. However, there are a number of issues with the presentation of results and analysis that need to be clarified and addressed. The followings are more specific comments for improving the manuscript:

1.      Lines 70-71
 The authors have indicated that the secretion of short-chain fatty acids (SCFAs) was induced by the administration of C. minuta. What are the SCFAs? Why there was no results shown in the study?

2.     Line 100: the Normal group (16% glycerol solution)
The reason of using 16% glycerol solution as the normal control should be supplemented in the Discussion section.  

3.     The amino acids metabolisms are very complicated in the biological systems. The study focused on the phenylalanine metabolism to convince the importance of the study. However, there was possible existed in a cross interaction with the other amino acids. The authors might be suggested to give the Discussion to convinced the key selection on the phenylalanine.

4.     Lines 244-
C.minuta should be revised to C. minuta.

Comments on the Quality of English Language

English language is fine.  Minor editing of English language would be required.

Reviewer 4 Report

Comments and Suggestions for Authors

Major Criticism

1: There is evidence that a change in the microbiome alters acetaminophen bio-disposition (PMID: 32165665).  Serum levels of acetaminophen would help support that the liver is seeing the same concentration of acetaminophen.

2:  The gavage solution introduces acetaminophen mixed in the same solution with the C. minuta bacteria.  There is evidence that acetaminophen can be degraded by some strains of bacteria or the mixing C. minuta with the drug can artifactually decrease the administered dose through uptake of drug into the probiotic.   A simple measure of remaining acetaminophen in the dosing solution after the bacteria is filtered out would show equivalent doses are administered between ± bacteria solutions. Many of the results can also be explained as a decrease in the acetaminophen dose.

3: Even daily administration of 109 C. minuta over 14 days is a tiny fraction of the total microbiome load.  The study included bacterial gene analysis of fecal material at the end of the study and the difference in the amount of C. minuta in control verse probiotic group is slight.  Are the differences significant? 

4. The discussion on the results is predicated on the assumption that probiotics are scientifically established to have implications on health and disease.  However, that is not established and is, in fact, quite controversial in regards to human health.  Due to the size of the industry and the amount of money it involves there is little desire in the nutraceutical industry to conduct rigorous trials. The addition of a paragraph in the discussion of how the study results can translate to the human is recommended, especially considering the results and many of the references supporting probiotic use are mouse based.

5.  The majority of acetaminophen liver toxicity is from overdose from suicide attempts.  How would the administration of a probiotic be of use in these situations?

Minor Criticism:

1.       Lines 106-108.  Were phenylalanine treated mice challenged with acetaminophen after 14 days?  There is no mention of that in the methods but Figure 5A suggests acetaminophen challenge.  This needs to be added to the methods.

2.       Line 170. Replace polyformaldehyde with paraformaldehyde.

3.       Line 518 – 522.  Need to direct reader to Figure 8H to support statement that atrazine metabolism is upregulated.  Also the statement is confusing.  Figure 8H suggests the phenylalanine group has much increase atrazine degradation which should decrease the amount of NAPQI generated from acetaminophen.  Figure 5D shows clear increase in ALT and AST with the phenylalanine group.  The inconsistency needs to be sorted out.

Round 2

Reviewer 2 Report

Comments and Suggestions for Authors

Authors correctly answered comments